

# Forecasting Carbon Monoxide on a Global Scale for the ATom-1 Aircraft Mission: Insights from Airborne and Satellite Observations and Modeling

Sarah A. Strode[1,2], Junhua Liu[1,2], Leslie Lait[2,3], Róisín Commane[4], Bruce Daube[4], Steven Wofsy[4], Austin Conaty[2,5], Paul Newman[2], Michael Prather[6]

[1]Universities Space Research Association, Columbia, MD, USA

[2]NASA GSFC, Greenbelt, MD, USA

[3]Morgan State University, Baltimore, MD, USA

[4]Harvard University, Cambridge, MA, USA

[5]SSAI, Greenbelt, MD, USA

[6]University of California, Irvine, CA, USA

*Correspondence to*: Sarah A. Strode (sarah.a.strode@nasa.gov)

**Abstract** GEOS-5 forecasts and analyses show considerable skill in predicting and simulating the CO distribution and the timing of CO enhancements observed during the ATom-1 aircraft mission. Using tagged tracers for CO, we find a dominant contribution from non-biomass burning sources along the ATom transects except over the tropical Atlantic, where African biomass burning makes a large contribution to the CO concentration. One of the goals of ATom is to provide a chemical climatology over the oceans, so it is important to consider whether August 2016 was representative of typical summer conditions. Using satellite observations of 700 hPa and column CO from the Measurement of Pollution in the Troposphere (MOPITT) instrument, 215 hPa CO from the Microwave Limb Sounder (MLS), and aerosol optical thickness from the Moderate Resolution Imaging Spectroradiometer (MODIS), we find that CO concentrations and aerosol optical thickness in Aug. 2016 were within the observed range of the satellite observations, but below the decadal median for many of the regions sampled. This suggests that the ATom-1 measurements may represent relatively clean but not exceptional conditions for lower tropospheric CO.



## 1 Introduction

The first phase of the NASA Atmospheric Tomography Mission (ATom-1) took place in July-August 2016. The aircraft completed a circuit beginning in Palmdale, California and traversing the remote Pacific and Atlantic oceans, providing an unprecedented picture of the chemical environment at a wide range of latitudes over the remote oceans. Chemical forecasts from the GEOS-5 model provided insight into the chemical environments and sources of pollution for the diverse regions sampled during the ATom-1 campaign.

ATom provides an observation-based climatology of important atmospheric constituents and their reactivity in the remote atmosphere. *Prather et al.* (2017) examined the ability of observations from a single path to represent the variability of a broader geographic region, but noted that year-to-year and El Nino/Southern Oscillation (ENSO) variability could also be important. Here we show how the time and place of ATom-1 measurements fit into a global, multi-year climatology of CO.

Year to year variability in meteorology and emissions both contribute to interannual variability in trace gases and aerosols. For example, ENSO is a major driver of variability in ozone distributions (*Ziemke and Chandra*, 2003), and large biomass burning events during El Nino years increase concentrations of trace gases including CO and $CO_2$ (*Langenfelds et al.*, 2002). Biomass burning plays a particularly strong role in driving the interannual variability of CO (e.g. *Novelli et al.*, 2003; *Kasischke et al.*, 2005; *Duncan and Logan*, 2008; *Strode and Pawson*, 2013; *Voulgarakis et al.*, 2015;). The impacts of large biomass burning events during El Nino events are visible in satellite observations of CO (e.g. *Edwards et al.*, 2004; *Edwards et al.*, 2006; *Logan et al.*, 2008, *Liu et al.*, 2013).

*Pfister et al.* (2010) used a chemistry transport model (CTM) as well as satellite data to examine the CO sources and transport over the Pacific during the INTEX-B mission compared to previous years. They found biomass burning to be the largest contributor to interannual variability, despite its lower emissions compared to fossil fuel sources.

In this study, we place the August 2016 ATom observations in the context of interannual variability and assess the contributions of different emission sources to the various regions sampled during the campaign. We focus on CO, a tracer of incomplete combustion whose lifetime of 1-2 months allows long-range transport to the remote oceans. Section 2 describes the model and observations used in this analysis. Section 3 compares the GEOS-5 CO to observations. Section 4 discusses the global distribution of CO, and presents the relative CO source contributions to the regions sampled by ATom. Section 5 presents an analysis of the interannual variability in CO and aerosol optical thickness seen in satellite observations to assess how well August 2016 observations represent climatological August conditions. Section 6 summarizes our conclusions.



## 2    Observations and Model

### 2.1    ATom Observations



ATom-1 flew transects through the Pacific, Southern, Atlantic and Arctic oceans with the NASA
DC8 aircraft in August 2016. Each of the 11 flights included sampling from the boundary layer to the top
of the aircraft range (39 kft). We use the ATom-1 data (July-August 2016) [*Atom Science Team,* 2017] for
comparison with the model forecasts and analyses.
We take ATom-1 CO observations from the Harvard QCLS instrument (*Santoni et al.*, 2014), which
has a history of successful measurements during the HIAPER Pole-to-Pole Observations (HIPPO)
campaign. Briefly, the instrument uses a pulsed quantum cascade laser at 2160 $cm^{-1}$ to measure absorption
of CO through a astigmatic multi-pass sample cell (with 76 m path length), with detection using liquid
nitrogen cooled HgCdTe detector. A separate laser and detector are used to measure methane and nitrous
oxide in the same cell.  Inflight calibrations were conducted with gases traceable to the NOAA WMO
(X2014) scale, with calibration of tanks before ATom1 and after ATom2 (February 2017) showing no
significant change in the CO concentration in the gas standards. The inlet for the instrument was specially
designed                for                the                DC-8                aircraft.

### 2.2    Satellite Observations


We use satellite observations that cover more than a decade to examine the interannual variability
of CO and aerosols.  The Measurement of Pollution in the Troposphere (MOPITT) instrument, which flies
on the Terra satellite, provides CO observations beginning in 2000 (*Edwards et al.*, 2004).  We use the
version 6 thermal infrared (TIR) level 3 product (*Deeter et al.,* 2014).  The MOPITT TIR averaging kernels
show high sensitivity to CO between 700 and 500 hPa (*Emmons et al.*, 2007).
The Microwave Limb Sounder (MLS) (*Waters et al.*, 2006), which flies on the Aura satellite,
provides useful observations of CO down to 215 hPa (*Livesey et al.,* 2008) beginning in 2004.  We use the
Version 4.2 level 2 data with the recommended quality, status, precision, and convergence criteria.
Although MLS data overlap with ATom only at the highest flight levels, both it and MOPITT provide
complementary views of CO in the lower troposphere and upper troposphere/lower stratosphere (UTLS),
respectively.
The Moderate Resolution Imaging Spectroradiometer (MODIS) instrument on the Aqua satellite
provides column aerosol optical thickness (AOT) data beginning in 2002.  We use the Collection-6 level 2
(MYD04_L2) [*Levy et al.*, 2015] 550 nm AOT data over oceans aggregated into 0.5 degree grid boxes, and
then take monthly means with the daily data weighted according to the QA.

### 2.3    Model Description


We use chemical forecasts and analyses from the GEOS-5 Forward Processing (FP) system.  The FP
stream from the Global Modeling and Assimilation Office (GMAO) generates GEOS-5 forecast products as



well as assimilation products using the most current system approved for near-real-time production. The
GEOS-5 model (*Molod et al.*, 2015) is a global general circulation model (GCM) with 72 vertical levels
reaching from the surface to 1 Pa. The assimilation system is described in (*Rienecker et al.*, 2008;
*Rienecker et al.,* 2011), and includes assimilation of ozone measurements from the Ozone Monitoring
Instrument (OMI) and MLS, and aerosol optical depth as well as meteorological variables. The forward
processing system produces output on 72 model levels or 42 pressure levels with 5/16 by 1/4 degree
horizontal resolution. Our study uses the pressure level output.

The GEOS-5 FP system (*Lucchesi*, 2017) simulates the transport of CO as well as tagged CO tracers

from specific regions and sources, which helps track the transport of pollution outflow. Tagged tracers are
available for biomass burning (BB) globally as well as biomass burning from Eurasia, North America,
Africa, and Central and South America; and for non-BB sources globally and from Europe, Asia, and North
America. Non-BB sources include fossil fuels, biofuels, CO from oxidation of biogenic VOCs, and CO
from methane oxidation, as described in *Ott et al*. (2010). *Bian et al*. (2013) used observations of
dichloromethane and acetonitrile from the ARCTAS mission to validate the anthropogenic and biomass
burning CO tracers, respectively.

Daily-varying biomass burning emissions come from the Quick Fire Emission Dataset (QFED)

version 2 [*Darmenov and da Silva*, 2015], which is based on fire radiative power from the MODIS
instrument. Thus the BB emissions include day-to-day and interannual variability, but the non-BB sources
and the OH fields use monthly means and lack daily-scale variability and interannual variability. Table 1
presents the August emission inputs for the major regions considered. CO emissions are then scaled up by
20% for fossil fuels and 11% for biomass burning to account for CO production from co-emitted VOCs.
CO from methane oxidation is included in the non-BB tagged tracers for the regions in which oxidation
occurred. The monthly mean methane fields come from a GMI Chemistry and Transport Model (CTM)
simulation, which uses prescribed zonal mean surface concentrations. CO is lost by reaction with OH
using fixed monthly OH fields archived from the GMI CTM. Supplemental Figure S1 shows the methane
and OH fields included in the FP system.

## 3    GEOS-5 Chemical Forecasting for ATom

During the ATom mission, the GEOS-5 model is engaged to provide chemical forecasts for each

flight that include the major chemical species and, for CO, tagged tracers for different sources. The
chemical forecasts are used together with meteorological forecasts for day-to-day flight planning, although
flight tracks were intentionally not altered to chase specific chemical features to avoid a highly biased
sampling of pollution.

We examine the performance of the GEOS-5 forecasts by comparing the simulated CO to the QCLS

observations. The forecasts provided during the mission used forecast wind fields, with the forecast lead
time varying depending on the timing of the flight. For consistency, the results shown here use the CO
simulated with the assimilated wind fields, but we note that similar features were seen for the CO simulated





with the forecast winds, as further discussed in section 3.1.2. For the model results, we do not apply
temporal interpolation between the model output frequency (6 hours). Instead, we sample the model
forecasts at the time closest to the mid-point of each flight segment. To compare with observations, the 3D
model forecast was interpolated to the longitude, latitude and pressure given in the 10-second merges of the
ATom measurements.

### 3.1     Analysis of CO along the Meridional Flight Tracks

We compare CO from GEOS-5 to the QCLS CO observations for specific flights, using the 10-
second merge files. The GEOS-5 CO is taken from the 3D field at the time closest to the mid-point of the
flight and interpolated in space to the flight track. We focus on two sections of the ATom-1 circuit: the
North to South flights through the Pacific, and the South to North flights through the Atlantic, although we
briefly discuss the other flights as well. These two transects allow us to examine the transition from
northern hemispheric to tropical to southern hemispheric influence.

### 3.1.1     Pacific legs

Figure 1 shows CO from the three Pacific flights spanning Anchorage, Alaska to Christchurch,
New Zealand. The top panels show the GEOS-5 curtain of CO along the flight track, with the QCLS
observations overplotted in circles. The observations show higher values of CO in the first half of the
Anchorage-Kona flight compared to the other portions of the Pacific, and this feature is reproduced in
GEOS-5 as well. GEOS-5 agrees well with the observed mean value for CO on this flight (Table 2).
Tagged tracers (Fig. 1 bottom panels) show that non-BB sources, especially from Asia, are the dominant
contributor to CO levels throughout the Pacific, and the decrease in Asian non-BB CO explains the
observed decrease in CO as the flights move south.
The observations show plumes of enhanced CO scattered throughout all three Pacific flights,
although they are most intense in the north Pacific, as seen in the Anchorage-Kona flight. GEOS-5
typically reproduces the timing of these plumes, but the magnitude is usually underestimated, particularly
for the strongest plumes. This leads to an underestimate of the observed standard deviation of the CO on
the Palmdale-Anchorage (Fig. S2) and Anchorage-Kona flights (Table 2). In addition to biases in
emissions, observations often show fine-scale structures too small for the model to resolve (Hsu et al.,
2004), and underestimating the concentrations in strong plumes is a common problem for global models
(e.g. *Heald et al.*, 2003). An exception is in the tropical Pacific (Kona-Pago Pago flight), in which GEOS-5
predicted some enhancements, driven by fossil fuels, not seen in the observations. Tagged tracers indicate
that Asian non-BB CO drove many of the observed enhancements, while others were due to biomass
burning.
In the south Pacific (Pago Pago to Christchurch segment), the flight sampled the stratosphere three
times, with CO levels decreasing to approximately 30 ppb, as shown in Fig. 1c. As expected from the basic
chemistry and seen in previous observations, both ATom-1 and GEOS-5 show a strong decrease in CO as
the flight rises above the tropopause, with GEOS-5 underestimating the observed decrease. Both the model



and measurements show tropospheric CO less than 90 ppbv along the flight route with slightly elevated CO
above 600 hPa around T22:00 and T24:00. For this flight and the subsequent flight to Punta Arenas, all
observations are in the Southern Hemisphere and the mean values for both ATom-1 and GEOS-5 agree
within the range 54-57 ppb (Table 2, Fig S3).

### 3.1.2 Atlantic Legs

The ATom flights traversed the Atlantic from South to North, beginning in Punta Arenas, Chile
and ending in Kangerlussuaq, Greenland. Figure 2 shows the Atlantic flights from Punta Arenas to
Ascension Island to the Azores to Kangerlussuaq. GEOS-5 has an excellent simulation of background CO
values seen on these flights, with the mean values falling within 2 ppb of the observations (Table 2) while
the mean observed values for each flight shift from 69 to 101 to 88 ppb. The observations show plumes of
high CO intersecting the flight track on all three flights. GEOS-5 also shows plumes of enhanced CO at
these locations, but the magnitude is often underestimated (Fig. 2d-f), especially for the Azores-
Kangerlussuaq flight. Supplemental Figure S4 shows the CO results for the Azores-Kangerlussuaq flight
using forecast wind fields, and illustrates the temporal evolution of CO plumes along the flight track.
Comparison of Fig. S4 with Fig. 2f shows the small impact of using analysis versus forecast wind fields.
Non-BB sources dominate the background CO levels on all three flights. However, biomass
burning plays a dominant role in the plumes of high CO (Fig. 2g-i). South American biomass burning
leads to CO enhancements between T14 and T16 of the Punta Arenas to Ascension flight. In the later
portion of that flight, biomass burning from Africa leads to strong CO plumes. Strong plumes of African
biomass burning are also seen at the beginning of the Ascension to Azores flight. GEOS-5 shows a strong
plume around 800 hPa for the first hour of the flight, which agrees well with observations (Fig. 2b,d). The
observations show additional strong plumes in the next hour between 600 and 700 hPa. These plumes are
present but underestimated in GEOS-5, possibly due to errors in the magnitude of the emissions or the
placement of the plumes.
The non-BB contribution to CO in the Atlantic reflects a mixture of global sources. Asian sources
make a notable contribution to the non-BB CO variability in the tropics (first half of the Ascension to
Azores flight), but as expected N. American sources become more dominant in the second half. In the
northern (later) portion of the Azores to Kangerlussuaq flight (Fig. 2), GEOS-5 attributes the observed
plumes to Eurasian biomass burning, but underestimates their magnitude. This flight also crosses the
tropopause, and both ATom-1 and GEOS-5 show a corresponding dip in CO concentrations. GEOS-5
predicts a plume of enhanced CO due to N. American emissions around 11Z of the Azores to
Kangerlussuaq flight that is not seen in the observations (Fig. 2f,i). A similar error is made in the
Kangerlussuaq to Minneapolis flight (Fig. S3). This could be due to either an error in the assumed N.
American sources, or to misplacement of the plume by the model. A large overestimate of CO at the end of
the Minneapolis to Palmdale flight also points to a potential error in North American emissions from either
fossil fuels or biomass burning.





## 3.2 Model Evaluation Summary

We summarize the comparison between the CO simulated by the GEOS-5 analyses and the QCLS observations in Figure 3. The majority of points lie near the one-to-one line, indicating good overall agreement between the GEOS-5 and observed CO distributions. The higher concentrations in the tropical Atlantic compared to the tropical Pacific are evident in both the observations and model. Fig. 3 also reveals occasional model overestimates of CO on flights over North America (green triangles), as well as underestimates of high CO plumes over the North Pacific and Tropical Atlantic. An underestimate of Eurasian biomass burning contributes to the model underestimates in the North Pacific and North Atlantic, and has implications for ozone production in aged BB plumes [*Liu et al*., in prep]. Globally, the correlation of simulated and observed CO with 5-minute binning is r=0.69. Correlations for the Pacific, Atlantic, and North America are 0.72, 0.80, and 0.80, respectively, while the correlation for the southern ocean is 0.053. The poor correlation for the southern ocean reflects the very low variability of CO in this region. The model performs far better at capturing the larger gradients present in the other regions. In general, the good agreement between model outputs and observations testify the model forecasting skill and suggest the suitability of using GEOS-5 forecast products to guide the design and execution of aircraft campaigns.

## 4 Source Contributions to the Global CO Distribution

### 4.1 Global CO Distribution

Figure 4 compares CO from GEOS-5 to the QCLS CO observations for the ATom-1 circuit including the 11 total flight segments. The GEOS-5 CO is taken from the analysis closest to the mid-point of the flight time and interpolated to the flight track following the longitude, latitude and pressure given in the observations. We average both model CO and ATom measurements into one point per 360-seconds for easier visualization.

Both model simulations and measurements show polluted air with higher CO mixing ratios in the northern hemisphere than that in the southern hemisphere in August 2016. Over the northern hemispheric polar region, the observations indicate highly polluted air with CO maxima occurring over Alaska and northwest Canada, features also seen in the GEOS-5 simulation. Over the Atlantic section, CO maxima with slightly lower values occur around the same latitude over west Greenland as shown both in observations and model simulation. CO over the northern most locations along the ATom-1 circuit see some low values both in model and observations, particularly north of 30°N and south of 40°S, due to the measurements occuring in the stratosphere or occurring in the upper troposphere with stratospheric influence. Both model and observations indicate that the air is relatively clean over the Pacific south of 30N with CO less than 70 ppb and the CO minimum around 60S over the southeast Pacific. Over the Atlantic section, both model and observations show low CO concentration south of 30S, but show a strong CO maximum over the tropical Atlantic (5S-5N) with CO greater than 120 ppb. This high CO is mainly driven by southern hemisphere BB. CO is slightly lower between 30N and 60N compared to that over tropical



Atlantic and the Greenland. The similarity between GEOS-5 and ATom-1 variability in neighboring points
is due in part to the vertical profiling which places horizontally extensive biomass burning layers in both
model and presumably the atmosphere at the same point along the track.
**4.2    CO Source Contributions**
We calculate the contribution of different CO sources to the total simulated CO using the GEOS-5
tagged CO tracers sampled along the ATom flight tracks. This analysis provides a picture of the dominant
sources affecting the constituent concentrations observed during ATom-1 for different regions of the
atmosphere. The tagging of CO sources includes both biomass burning (BB) and non-biomass burning
(non-BB) from four continental areas, with all other sources put into the "other" bin. Other BB sources are
small, but other non-BB sources are quite large as they include all natural sources as well as atmospheric
photochemical sources such as methane oxidation.
Figure 5 shows the contribution of each tagged tracer over the Pacific Ocean from 120°E to 110°W,
averaged over 5 degree latitude bins. Non-biomass burning sources dominate at all latitudes, due in part to
the inclusion of CO from methane oxidation in addition to fossil fuel sources in these tracers. The
oxidation of methane over the remote oceans contributes to the large magnitude of "other non-BB" sources
over the southern latitudes of the Pacific. Asian non-BB sources make the largest contribution to middle
and upper tropospheric CO (Fig. 5a) at the mid-latitudes of the North Pacific, with smaller contributions
from N. American and European non-BB sources. The largest biomass burning contribution comes from
Africa in the Southern Hemisphere and Tropics, switching to Eurasia in the northern latitudes.
Figure 5b shows the relative contributions in the lower troposphere, including the marine boundary
layer and defined here as pressures greater than 850 hPa. Missing bars indicate latitudes where no ATom-1
measurements were made in the lower troposphere. Asian non-BB CO makes a smaller contribution in the
lower troposphere than in the middle and upper troposphere. A strong CO maximum around 30°N is more
pronounced in the lower troposphere than above. This bin is not representative of the remote Pacific as it
includes Palmdale, California, with large contributions from local North American BB and non-BB
sources.
The Atlantic flights (0°-60°W) show a large contribution from other non-BB sources in the
Southern Hemisphere with increasing contributions from Asian, N. American, and European non-BB CO
as the flight moves northward (Fig. 6), similar to the picture over the Pacific. However, the Atlantic
receives a larger contribution from biomass burning, particularly from Africa, over the Tropics. The
contribution from African BB is strong throughout the troposphere, but is particularly pronounced in the
lower troposphere, where it exceeds 100 ppb in the bins centered at 10°S and 5°N.
We also examine the tagged tracer contributions for each flight, including all altitudes sampled by
the flight (Fig. 7, Supp. Table S1). Flights occurring in the tropics and southern hemisphere (Flt. 1, 4-8)
receive 44-75% of the total CO from other non-BB sources. Other non-biomass burning sources include all
non-biomass burning sources located outside North America, Europe, and Asia. The contribution from



methane oxidation in addition to southern hemisphere emissions explains this large contribution.  Flight 8
has a somewhat lower percent contribution from other non-BB sources than the other southern hemisphere
and tropical flights due to the higher percent contribution from African biomass burning.  In contrast, the
Northern Hemisphere flights have a larger contribution from northern hemisphere source regions.  Asian
non-BB explains over a third of the total CO for the northern Pacific flights (Flt. 2-3), while Asian and N.
American non-BB sources make comparable contributions to the North Atlantic and N. American flights
(Flt. 9-11).

Since Figs. 5 and 6 reveal differences in source contributions between the lower troposphere and the

middle and upper troposphere, we also examine the source contributions to each flight for the lower
troposphere (Pressure > 850 hPa) only (Supp. Fig. S5).  Asian sources make a larger percent contribution to
the Pacific flights (Flt. 0-4) when all flight altitudes are considered rather than the lower troposphere alone.
Regional sources such as African biomass burning for flights 6 and 7 and N. American sources for Flights 9
and 10 make a larger percent contribution in the lower troposphere.
**5     August 2016 in the Context of Interannual Variability (IAV)**

One of the major goals for the ATom campaign is to produce a climatology based on un-biased,

representative samples (*Prather et al.*, 2017).  It is therefore important to consider whether August 2016 is
a "typical" summer month.   We focus here on the temporal representativeness of the ATom-1 campaign.
Spatial representativeness is investigated in *Liu et al.* [in prep].  August of 2016 was ENSO neutral, with a
multivariate ENSO index (MEI) (*Wolter and Timlin*, 1993); (https://www.esrl.noaa.gov/psd/enso/mei) of
0.175 for July/August.   However, it was preceded by strong El Nino conditions in 2015 and early 2016
(*Blunden and Arndt*, 2016).   We therefore consider whether the CO concentrations in August 2016 are
typical or anomalous.

Multi-year satellite records provide a valuable tool for determining how CO concentrations in the

regions of the ATom-1 flights compare to previous years.   We focus our analysis of CO interannual
variability on several regions traversed by the ATom flights.  Figure 8 shows these regions in black squares
overplotted on the MOPITT CO column for August 2016.  We also examine the IAV in BB sources from
nearby regions, outlined in red on Fig. 8.  Figure 9 shows box-and-whisker plots of the mean, minimum,
$25^{th}$, $50^{th}$, and $75^{th}$ percentiles, and maximum in monthly mean August CO for each region over the 2000-
2016 period for MOPITT (CO column and CO at 700 hPa) and 2004-2016 for MLS (CO at 215 hPa).  The
corresponding time series are shown in Fig. 10.  The variability in CO BB emissions from GFEDv4 for
2000-2016 is also shown for BB regions that may affect the ATom flights.  The BB emissions are averaged
over June through August to account for the persistence of CO in the atmosphere.

Among the regions mapped here, the tropical Atlantic shows the highest average CO values, as

well as the highest 2016 CO values, in both MOPITT and MLS observations (Fig. 9).  This is consistent
with large biomass burning emissions from southern hemisphere (SH) Africa transported into the tropical
Atlantic.  While SH Africa has the largest magnitude of biomass burning, its relative variability is smaller



than for the other regions (Fig. 9b).  Similarly, the IAV in the MOPITT CO column and 700 hPa level over
tropical Atlantic is smaller than that of the North Atlantic and Alaska regions.  Although the variability of
CO over tropical Atlantic is relatively small, the MOPPIT CO column shows a statistically significant anti-
correlation between the MOPITT CO column over the tropical Atlantic and the MEI (r=-0.52).  This
relationship is not significant for the MOPITT 700 hPa level.
The time series of August MOPITT CO columns for both the North Atlantic and Alaska, regions
that show high variability, show a small but significant temporal correlation with summertime Siberian
biomass burning (r=0.52 for the North Atlantic and r=0.59 for Alaska).  Slightly lower values are seen for
the 700 hPa MOPITT level.  The time series of August MOPITT 700 hPa CO shows an increase in 2003
for the North Atlantic and in 2002 and 2003 for Alaska (Fig. 10b,c).  Previous studies attribute peaks in
these years to the presence of large forest fires in western Russia and Siberia, respectively (Edwards et al.,
2004;Yashiro et al., 2009;van der Werf et al., 2006). MOPITT CO values were below average in 2016 for
both the North Atlantic and Alaska even though Siberian biomass burning was above average in 2016 (Fig.
9a, b).
Since ENSO is known to drive large biomass burning variability in Indonesia (*van der Werf et al.*,
2006), we consider whether it may influence CO concentrations over the New Zealand region.  Although
the MOPITT CO column over the Indonesia region does correlate with the MEI (r=0.64), there is no
significant correlation between June-Aug biomass burning in Indonesia and MOPITT CO over New
Zealand.  However, August is not the peak season for Indonesian biomass burning (*Duncan et al.*, 2003b).
The large Indonesian fires that occurred during the strong 1997/1998 El Nino peaked during September to
November (*Duncan et al.*, 2003a) and active fire detections for the 2015 Indonesian fires peaked in
September and October (*Field et al.*, 2016).  Thus we might expect Indonesian biomass burning variability
to have a greater influence on CO variability during the autumn season, which was sampled in ATom-3.
How does 2016 compare to previous years?  The MOPITT CO column shows tropical Atlantic CO
was near the 75[th] percentile, while the 700 hPa MOPITT level shows it close to the median.  This difference
arises because the MOPITT column also includes information from the upper troposphere, and the
MOPITT 200 hPa level (not shown) suggests CO levels for 2016 were near the 75[th] percentile.  In contrast,
MLS shows that 2016 CO in the upper troposphere was much lower than average, near the 25[th] percentile.
The MOPITT v6 TIR product has a small positive bias drift in the upper troposphere of 0.78 % yr-1 for the
200 hPa level (*Deeter et al.*, 2014), which may contribute to the higher rank of 2016 in the MOPITT upper
tropospheric data compared to MLS.  It is therefore hard to argue that 2016 was outside of the normal IAV
for this region.
2016 CO in the North Atlantic and Alaska regions was below average in both the MOPITT
column and the 700 hPa level, and is in fact the lowest August value in the MOPITT record for the 700 hPa
level over Alaska.  MLS also shows moderately low CO in the upper troposphere over Alaska in Aug.
2016.  Combined, this data suggests that the ATom-1 CO is not typical for the region.  August 2016 CO
column values are also below the median over New Zealand and the eastern and central tropical Pacific, but




the relatively low variability of these regions makes this less of a concern for the representativeness of the
ATom measurements. The IAV of these regions is larger for the MOPITT 700 hPa level, and 2016 lies
slightly below the $25^{th}$ percentile for this level.
The regionally-averaged 500 nm AOT from MODIS (Fig. 11) shows similar features to the
MOPITT column. The highest values are found for the tropical Atlantic, followed by the Alaska and North
Atlantic regions. However, the difference between the tropical Atlantic and the other regions is larger in
the aerosol case, while the difference between the North Atlantic and the Pacific regions is smaller. There
is also greater relative year-to-year variability over the tropical Atlantic for the aerosols than for CO. The
shorter lifetime of aerosols compared to CO, as well as the large contribution from biomass burning, likely
explains the greater prominence of the tropical Atlantic in the aerosol case. Furthermore, AOD (Fig. 12)
shows a clear peak in 2009 in several of the regions, whereas MOPITT data is missing for Aug. 2009, but
MLS shows a minimum (Tropical Atlantic) or no anomaly (other regions).
In summary, the multi-year satellite record shows considerable variability in CO, particularly over the
North Atlantic and Alaska. Concentrations during August 2016 were on the low end of the distribution for
most regions, especially in the lower troposphere. *Worden et al.* (2013) showed negative trends in the
MOPITT CO column significant at the one-sigma level for both the Northern and Southern hemispheres for
2000-2012. In addition, *Deeter et al.* (2014) report a small negative bias drift in the MOPITT V6 TIR
product in the lower troposphere, although drift in the column is almost negligible. Decreasing MOPITT
CO over time is also visible in some regions in Fig. 10. This negative trend may be contributing to the low
values in 2016. Overall, the year 2016 shows anomalies for some regions, but does not appear to be an
extreme year.
**6    Conclusions**
We place the observations from the ATom-1 campaign in the context of interannual variability and
global source distributions using satellite observations and tagged tracers from GEOS-5, respectively.
GEOS-5 gives a reasonable reproduction of the background CO levels for most flights despite the use of
climatological fossil fuel and biofuel emissions, and captures the global distribution of CO observed during
ATom-1. Simulations with both forecast and analysis winds capture the timing of many of the enhanced
CO plumes encountered during the flights, although the magnitude of the enhancements was often
underestimated, which is not unexpected given the difference in resolution between the observations and
model. The strong performance of GEOS-5 with regards to the overall CO distribution and the timing of
the enhancements gives us confidence in using tagged tracers to identify the sources affecting the air
sampled in ATom-1.
We find that for most flights the dominant contribution to total CO is from non-biomass burning
sources, which include both fossil fuels and biofuels and oxidation of hydrocarbons including methane. An
exception to this is in the lower troposphere of the tropical Atlantic, where biomass burning from Africa
makes the largest contribution, exceeding 100 ppb in some locations. The non-BB source includes a large





fraction from Asia for flights over the North Pacific and from both Asia and North America for the North Atlantic and North American flights, while other regions dominate in the Southern Hemisphere. Plumes of elevated CO from both biomass burning and non-BB sources led to observations of enhanced CO during ATom-1.

We use satellite observations of CO from MOPITT and MLS and AOT from MODIS to assess whether August 2016, the period sampled by ATom-1, is typical or atypical in the context of IAV in the satellite record (2000-2016). MOPITT and MLS show that CO in the lower and upper troposphere, respectively, were below average in August 2016 compared to the satellite record for August for most of the regions sampled by ATom-1, but not usually the minimum year. CO concentrations in the North Atlantic and Alaskan regions show a positive correlation with Siberian biomass burning and large interannual variability. In contrast, both MODIS AOT and the MOPITT CO column show above average values for the Tropical Atlantic in 2016. This suggests that the high values of CO and aerosols from biomass burning encountered during the tropical Atlantic portions of ATom may have been especially pronounced during this particular year.

The seasonality of biomass burning, the OH distribution, and atmospheric transport pathways can alter the source contributions from season to season. Thus, the next three ATom campaigns, which occur in different seasons, will likely show variations in the relative source contributions to each region.

**Data Availability**

Data from ATom-1 is available on the ESPO archive (https://espo.nasa.gov/home/atom/archive/browse/atom). MOPITT data is available at https://eosweb.larc.nasa.gov/datapool. MLS data is available from https://mls.jpl.nasa.gov/. MODIS aerosol data are available from https://ladsweb.modaps.eosdis.nasa.gov/api/v1/productGroupPage/name=aerosol.

**Acknowledgements**

The authors thank the NASA GMAO for providing the GEOS-5 forecasts and analyses. We thank the NASA Earth Venture Suborbital Program, ESPO, and the pilots, crew and support staff of the DC-8.

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



**Table 1 : Regional August 2016 CO Emission Totals in the GEOS-5 FP Simulations**

|                  | Fossil Fuel[1] | Biogenic[1] | BB[1] |
|------------------|:----------:|:--------:|:----:|
| North America    | 6.7        | 5.8      | 2.4  |
| Europe           | 4.9        | 2.4      |      |
| Asia             | 26         | 7.9      |      |
| Eurasia[2]       |            |          | 3.0  |
| Africa           |            |          | 24   |
| South America    | 13         | 17       | 11   |
| Other[3]         |            |          | 2.9  |
| Global           | 50         | 34       | 43   |

[1]Emissions are in units of Tg.

[2]The Eurasian tagged tracer for BB CO includes emissions from Europe and northern Asia, but excludes
southern Asia.

[3]Other fossil fuel emissions includes emissions from Africa and South America, while other BB emissions
excludes those regions since they are tagged separately.

**Table 2: Mean and Standard Deviations in CO along Atlantic and Pacific Flight Tracks**

| Region | Flight | Obs Mean (ppb) | Obs Stdev (ppb) | Model Mean (ppb) | Model Stdev (ppb) |
|--------|--------|:-----:|:-----:|:-----:|:-----:|
| Eastern Pacific | 1. Palmdale – Palmdale | 75 | 14 | 77 | 19 |
|  | 2. Palmdale - Anchorage | 100 | 40 | 88 | 16 |
| Pacific | 3. Anchorage-Kona | 85 | 36 | 81 | 18 |
|  | 4. Kona –Pago Pago | 61 | 5.1 | 63 | 5.5 |
|  | 5. Pago Pago – Christchurch | 55 | 11 | 57 | 6.1 |
| Southern Ocean | 6. Christchurch – Punta Arenas | 56 | 6.4 | 54 | 4.7 |
| Atlantic | 7. Punta Arenas – Ascension | 69 | 17 | 71 | 26 |
|  | 8. Ascension – Azores | 101 | 36 | 103 | 27 |
|  | 9. Azores – Kangerlussuaq | 88 | 32 | 87 | 19 |



| N. America | 10. Kangerlussuaq – Minneapolis | 90 | 26 | 91 | 22 |
|---|---|---|---|---|---|
| | 11. Minneapolis – Palmdale | 84 | 38 | 107 | 78 |




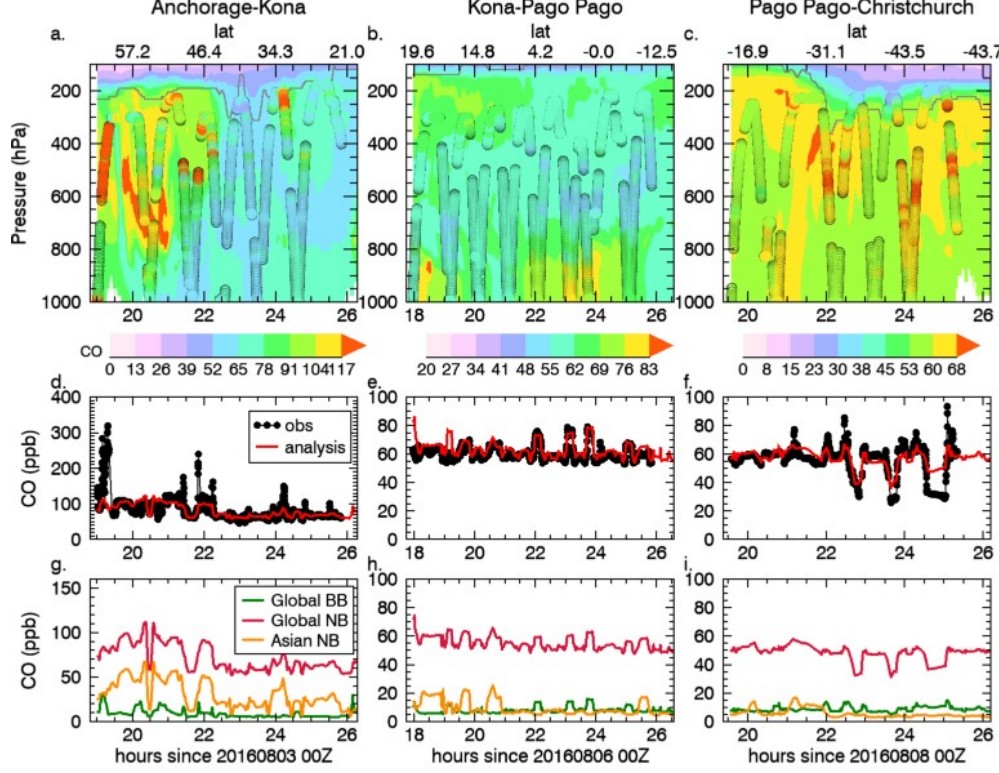


**Figure 1: Curtain plot of CO (ppb) from the GEOS-5 analysis as a function of time and pressure overplotted with the model tropopause (gray line) and QCLS CO observations (circles) (top row) for the a) Anchorage to Kona flight, b) Kona to Pago Pago flight, and c) Pago Pago to Christchurch flight. Axis ranges vary between panels due to the large range of concentrations encountered. The top x-axis indicates the latitudes of the flight track. d-f) The GEOS-5 CO interpolated to the flight track (red line) is compared to the observations (black circles). g-h) Tagged tracer contributions to the GEOS-5 CO.**



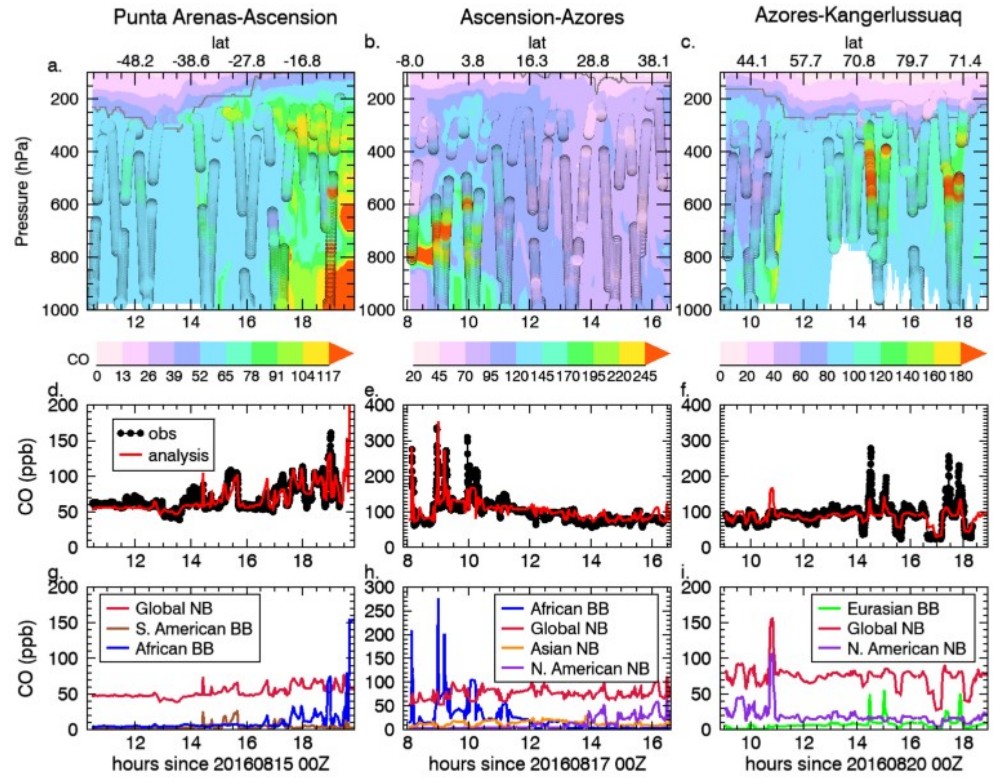

563

**Figure 2: As in Fig. 1, but for the Atlantic flights: a,d,g) Punta Arenas-Ascension Island, b,e,h) Ascension Island to the Azores, and c,f,i) Azores to Kangerlussuaq.**



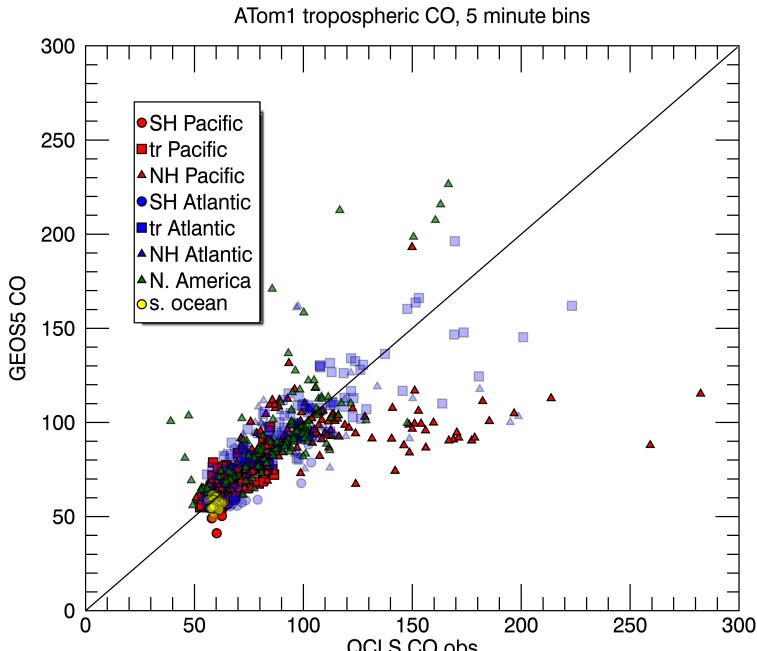

566

**Figure 3: GEOS-5 simulated CO versus QCLS CO observations for all ATom-1 flights averaged into 5 minute
bins. CO is in units of ppb. Pacific flights are shown in red, Atlantic flights in blue, N. American flights in
green, and southern ocean flights in yellow. Circles indicate Southern Hemisphere points, triangles indicate
Northern Hemisphere points, and squares indicate tropical points. The one-to-one line is overplotted in black.**

571





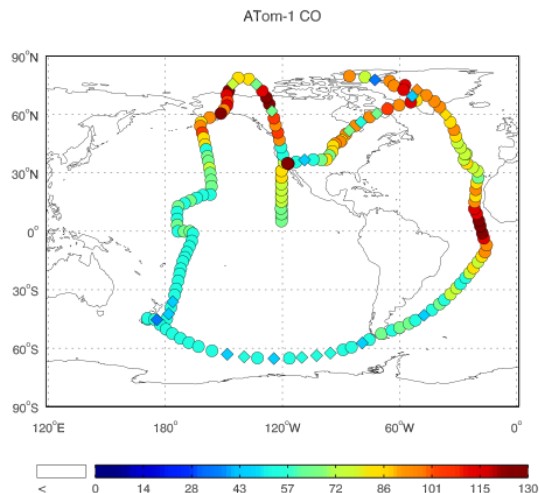

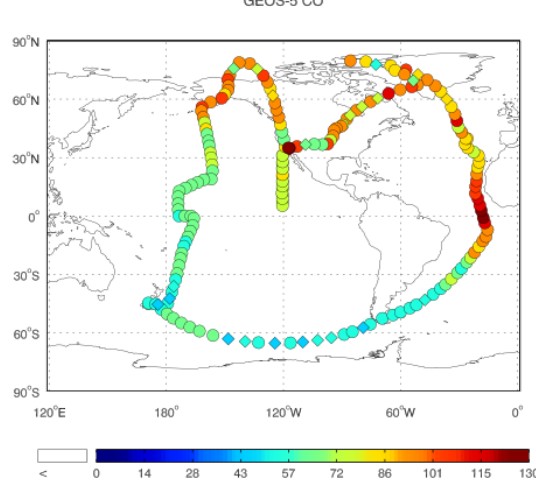

572

**Figure 4: CO (ppb) from the QCLS observations (top) and GEOS-5 analysis (bottom) for the ATom-1 circuit including all 11 research flight segments. The GEOS-5 CO is taken from the analysis closest to the mid-point of the flight time and interpolated to the flight track following the longitude, latitude and pressure given in the observations. Both model forecast and ATom measurements are averaged into a sample rate of one per 360-second. Data in the troposphere are plotted in a circle, while data in the stratosphere are plotted in a diamond, based on the GEOS-5 tropopause.**

579

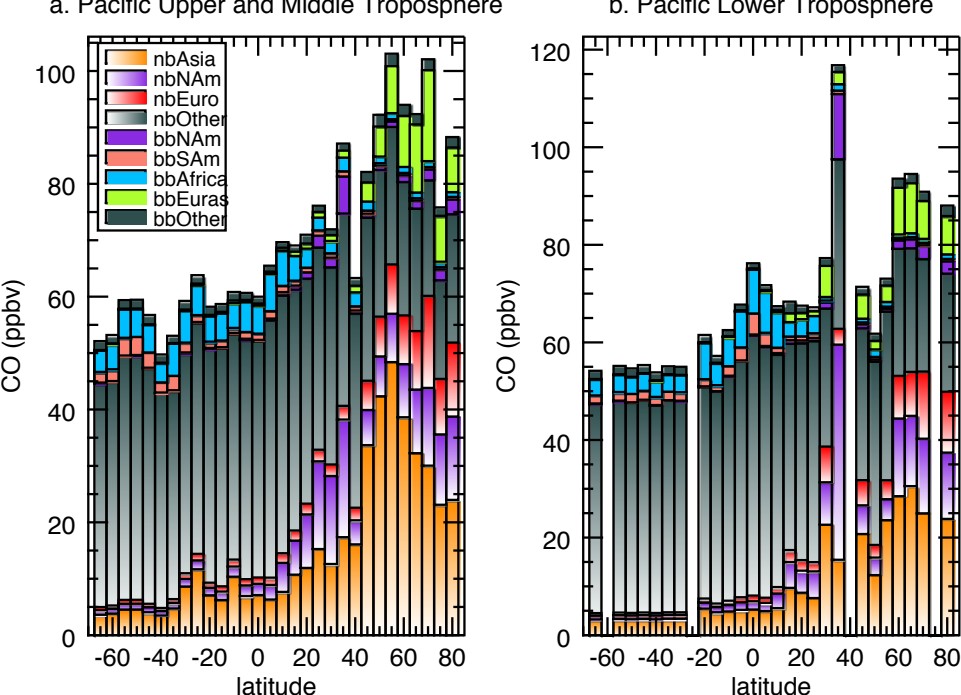

580

**Figure 5: The contribution of each tagged CO tracer over the Pacific in the (a) upper and middle troposphere**
**(pressure <= 850 hPa) and (b) lower troposphere (pressure > 850 hPa). Data from multiple flights over the**
**region between 120°E and 110°W is included, with each bar representing data averaged over a 5 degree latitude**
**bin. Shaded bars represent non-BB CO from Asia (orange), N. America (purple), Europe (red), and the rest of**
**the world (gray). Solid bars represent BB CO from N. America (purple), S. America (pink), Africa (cyan),**
**Eurasia (green), and the rest of the world (gray).**





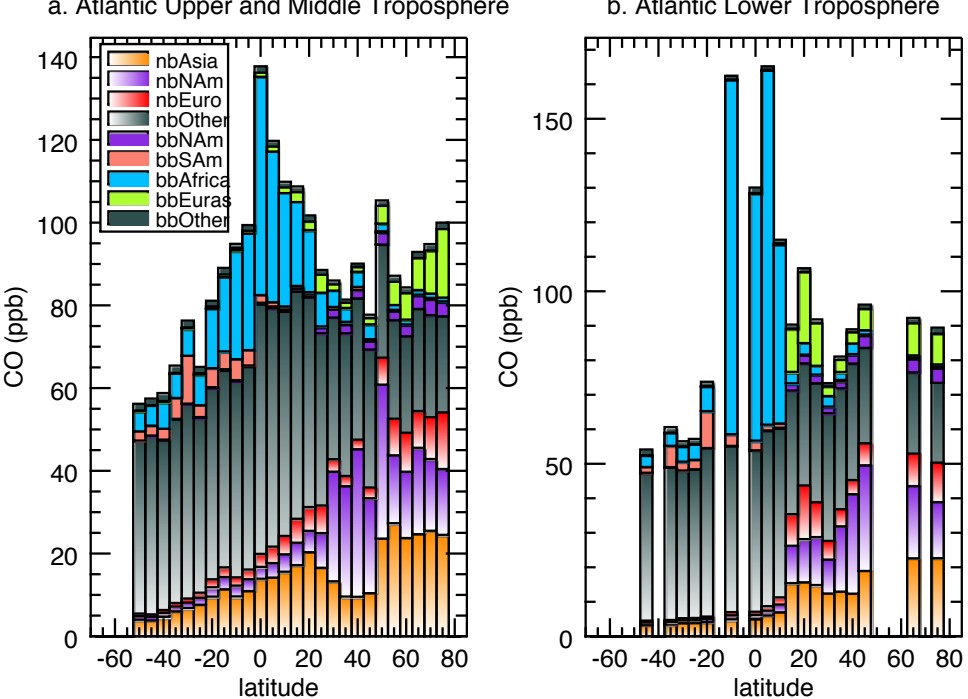


**Figure 6: As in Fig. 5, but for the Atlantic. Data from multiple flights over the region 0-60°W is included, with each bar representing data averaged over a 5 degree latitude bin.**




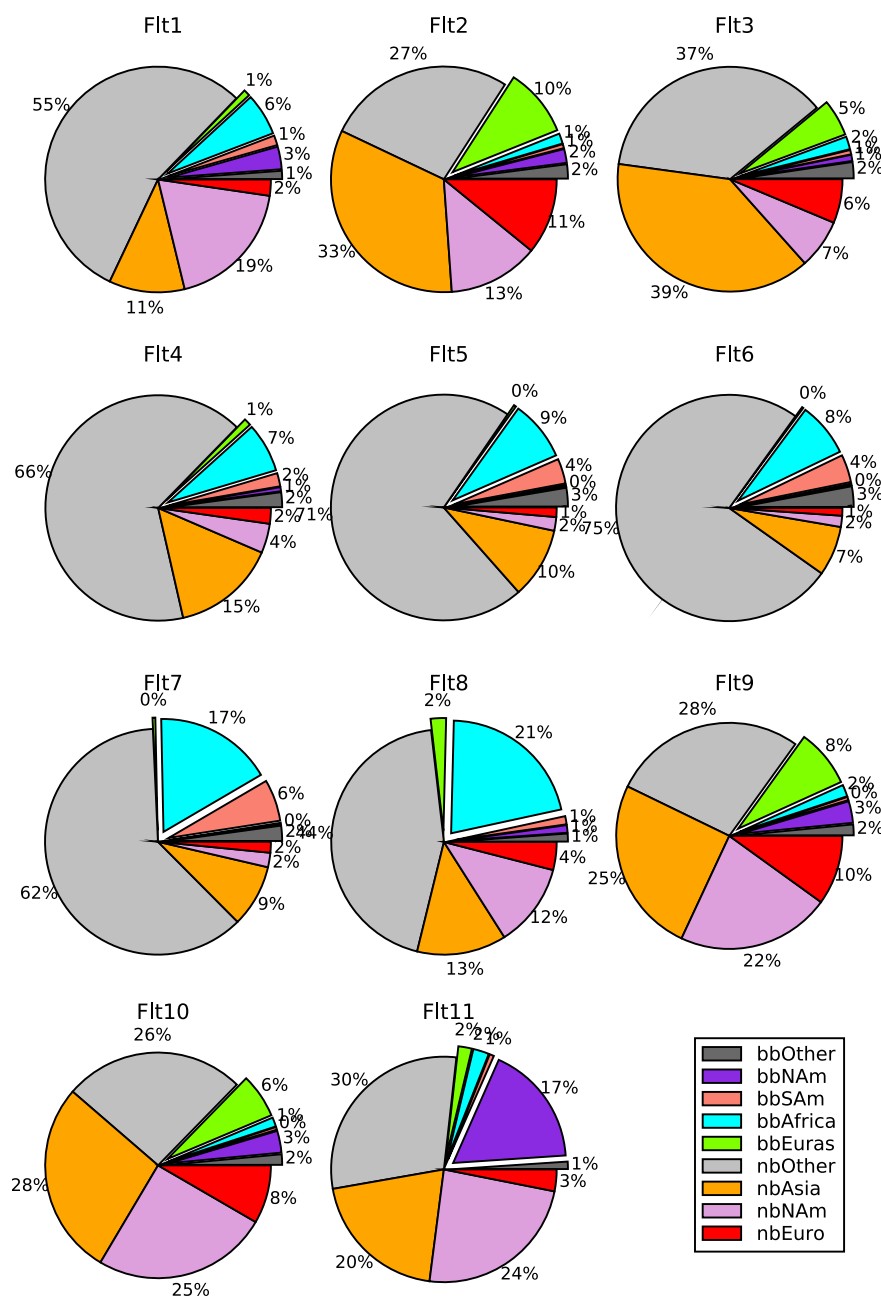

**Figure 7: Percent contributions of tagged tracers to total CO for each flight. Exploded slices represent the**
**biomass burning tracers: North American (purple), S. American (salmon), African (cyan), Eurasian (green), and**
**Other (dark gray).  The non-biomass burning (nb) tracers are for Asia (orange), N. America (lavender), Europe**
**(red), and other (light gray).**


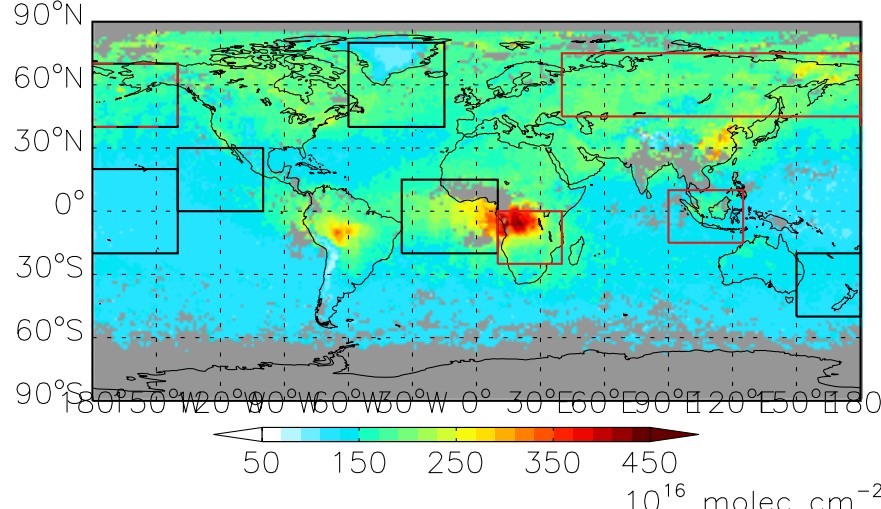

**Figure 8: MOPITT CO column for August 2016 overplotted with the regions shown in Fig. 10.  Black rectangles**
**indicate the regions where we analyze CO concentrations, and red rectangles indicate the regions used for**
**biomass burning.**



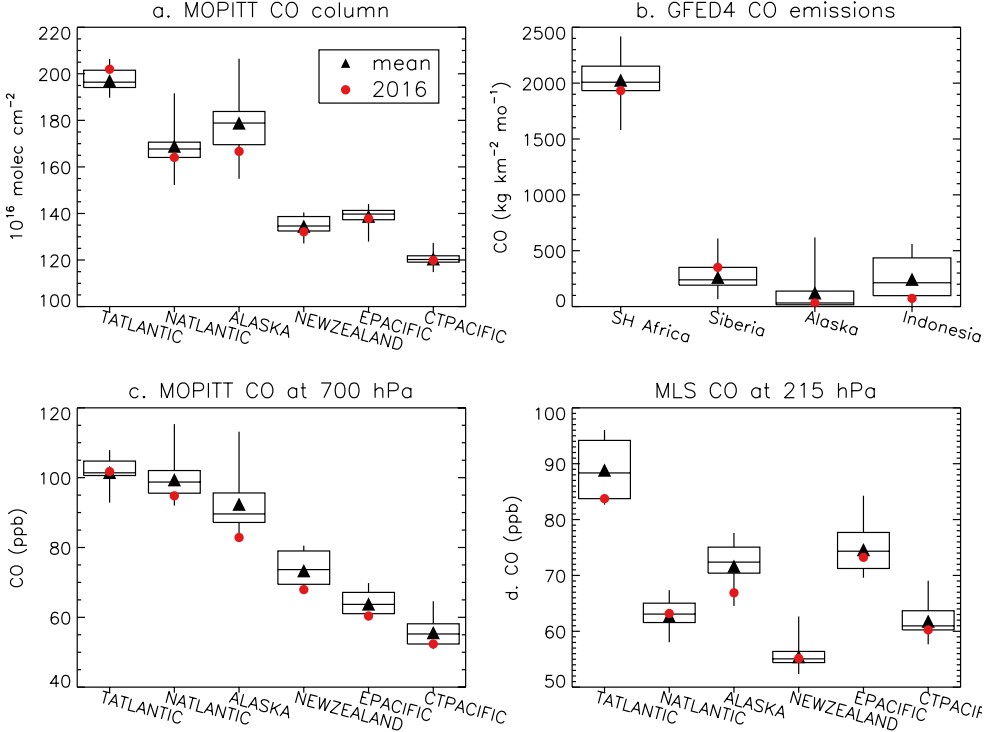


Figure 9: Boxes show the 25th, 50th, and 75th percentile values; whiskers show the minimum and maximum
values; black triangles show the mean value, and red circles show the 2016 value for a) the MOPITT CO
column, b) the GFED4 CO emissions, c) MOPITT CO at 700 hPa, and d) MLS CO at 215 hPa. Statistics for
MOPITT are for 2000-2016, statistics for GFED4 are for 2000-2015, and statistics for MLS are for 2004-2016.
MOPITT and MLS values are for August, while the GFED4 emissions are averaged over June through August.





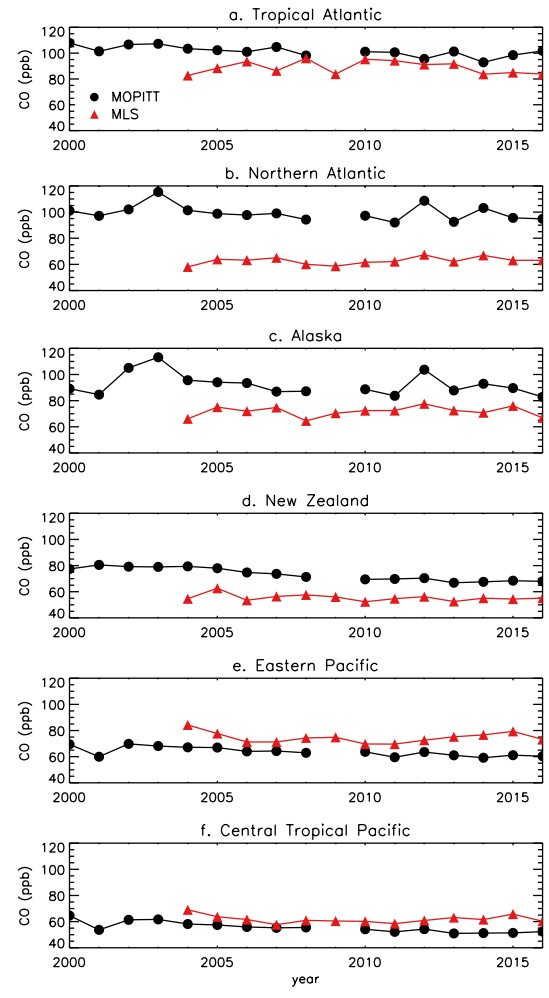



**Figure 10: Time series of August MOPITT CO at the 700 hPa level (black circles) and MLS CO (red triangles)**
**at the 215 hPa level for the 6 regions shown in black in Fig. 8.**

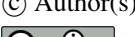



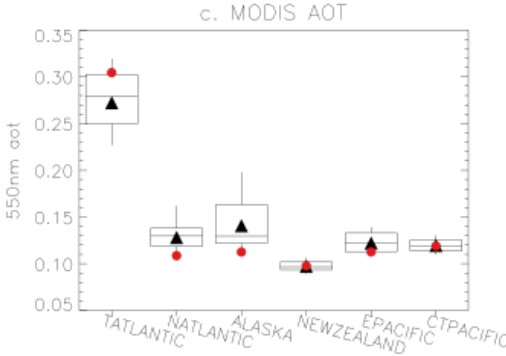


**Figure 11: As in Figure 9, but for the MODIS 550 nm AOT. Only values over oceans are included in the regional averages.**

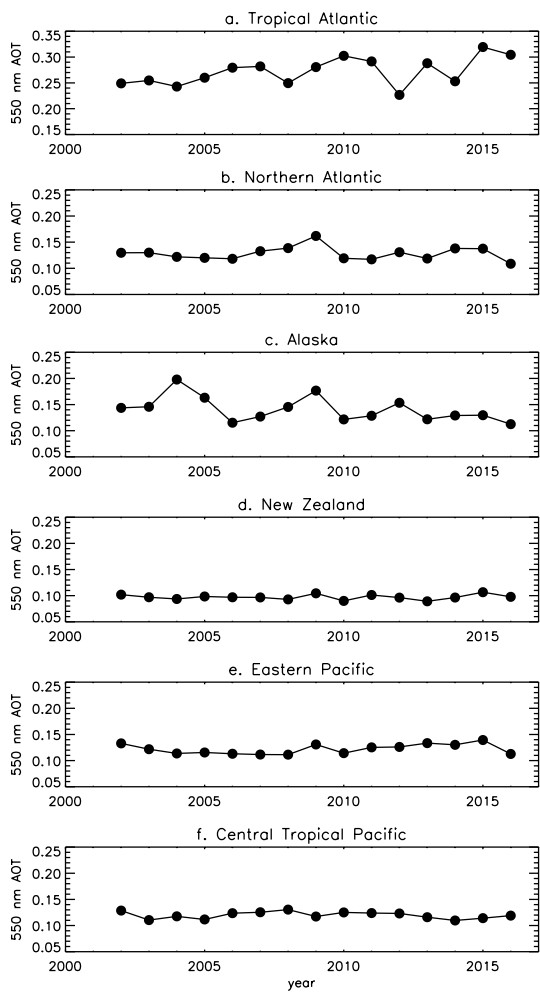

**Figure 12: Time series of regionally averaged MODIS 550 nm AOT. Only values over oceans are included in the**



**regional averages. The y-axis range for panel a differs from the other panels due to the higher AOT values in**
**that**                                                      **region.**