# Peer review of "ATom-1 Aircraft Mission: Insights from Airborne and"

_Atmospheric Chemistry and Physics, 2018_

## Referee Comment (RC1) · Anonymous Referee #1 · 18 Apr 2018

The authors use the GEOS-5 chemical forecast system to simulate the CO distribution and the timing of CO enhancements observed during the Atom-1 aircraft mission in July-August 2016, using tagged CO tracers to attribute modelled CO to non-biomass burning and biomass burning (BB) sources from different regions. The authors also use multi-year satellite measurements (MOPITT and MODIS) of CO and AOD to verify if August 2016 is representative of typical boreal summer conditions. One of the goals of Atom is to derive a chemical climatology over the ocean. The paper is well written and the authors have conveyed their results and findings clearly. The methods used

to derive the results are sound and thorough. I recommend this paper to be published with some clarification and improvements, detailed below.

Specific comments

1. My major concern is that the authors combined BB emissions from Europe and from Northern Asia into one tagged CO tracer, which I think leads to difficult attributions of the simulated North Atlantic CO to emissions in these two regions. One would suspect that European BB emissions have a more direct impact on North Atlantic CO peaks than the Asian BB emissions. Moreover, the text seems to imply that BB emissions from Southern Asia and Australia are not included in the simulation, which might lead to the underestimation of simulated South Pacific CO peaks.

2. Can you add to the map showing the Atom flight route the names of locations mentioned in the text and the labels of flight numbers?

3. L116-118: How do you justify adding 20% fossil fuel and 11% BB to CO emissions? What is the basis, and do you have a reference for the scaling factors? Please also show the effective CO total emissions, after scaling, in Table 1.

4. L119-120: Which regions? Methane is well-mixed, and is oxidised in all regions. Could you clarify?

5. L152-154: Could you elaborate why the observed peaks in Figure 1d are not captured by the GEOS-5 analysis? Could it be due to the underestimation of Asian BB or/and non-BB in the inventory?

6. Did you include biomass burning from South Asia and Australia in your CO simulation? If these are not included, would this lead to the underestimation of the peaks in observed CO in Figure 1f? This is related to the comment above.

7. L194-196: It is somewhat surprising that this region has notable Asian non-BB influence; could it be the case that the majority of this influence originates from Europe or Central Asia rather than from the Far East?
8. L197-198 and Figure 2f: Would it more precise to attribute the observed CO plumes here to European BB? Could N. American BB contribute to the plumes? Separating European and Asian BB emissions could shed more light on this.

9. L212-213: "An underestimate of Eurasian biomass burning contributes to the model underestimates in the North Pacific and North Atlantic, . . ." – same comment as above.

10. Figures 5 and 6: I suggest to use the same scale between the 2 panels in each figure.

11. L364-369: The negative trends are not necessarily associated with the lower than average value of 2016. Often, IAV is larger than the trend. Whether 2016 is extreme or not should be defined based on statistical metrics.

Technical corrections

1. L22: add "boreal" before "summer"

2. L293: replace "summer" with "boreal summer/austral winter"

3. L335: replace "autumn" with "September to November"

4. Figure 8: please tidy up the label on the x-axis.

5. Figures 11 and 12: please add "August" in the caption.

6. Table S1: flights are labelled from 0 to 10 – should it be 1-11?
* * *

---

## Referee Comment (RC2) · Anonymous Referee #2 · 19 Apr 2018

General Comments

A very unique set of measurements obtained during the ATom-1 mission was analyzed throughout this study. Comprehensive data sets, both from in-situ and satellites, and a state-of-the-art global chemistry model were utilized. The analyses methods used here were reasonable and the results are presented in logical manner. However, it is not clear to me that what the scientific backgrounds are and what scientific questions the authors try to answer. What new and unique findings are discussed in this study? What can we learn from the results presented here? Specifically, why is it important

for GEOS-5 to be able to simulate and forecast CO over the ocean? And why ATom-1 (and also the future deployment phases) was planned and how is it different from the previous campaigns, for instance, HIPPO? I think including more scientific background and contexts, and specific goals will make this article far more interesting.

Specific Comments

P1 (Abstract) – I think the abstract can be rewritten in order for it to serve as a brief summary of this study. The simplest way is to rearrange it. 1) What is the goal of ATom-1, 2) What are the observational findings shown in this study, 3) Why is the modeling study performed and 4) What the conclusion is.

P2 (Introduction) – This section can be rewritten to add clarity as well. It should include 1) Scientific background and goals of the ATom-1 (and other phases) mission, 2) What are the scientific questions ATom-1 aimed to answer? 3) What did ATom-1 sample and accomplish? 4) What is the goal of this study? In addition, the information in the third and the fourth paragraphs (lines 42-53) provides general overview, instead of specific to this study. I would also like to know what the ATom-1 specific references are. Is Prather et al. (2017) the only reference relevant to ATom-1?

Technical Comments

P3 (Section 2.1) – What are the vertical ranges of ATom-1 measurements? The accuracy or precision of the species measured during ATom-1 should be provided here.

P3 (Section 2.2) – The vertical range of MOPITT and MLS and their overlap have to be mentioned specifically. What pressure ranges do they overlap? How often did ATom-1 sample stratosphere? Are there any ground-based measurements to compare? It has to be mentioned why only the satellite data are used in the comparison. Also, why was the MODIS AOT used?

P3 (MOPITT) – I believe the current version of MOPITT is 7 not 6.

P4 (Section 2.3) – It has to be mentioned why the model is used in this study. What are

the questions that can be answered through modeling? Is a global model the sufficient tool for this study?

P4 (L125) – What could be learned from the chemical forecasts during a field campaign, such as, ATom-1? How many pollution or non-pollution events occurred during the ATom-1 period?

P5 (L160) – Is the bias in the emissions responsible for the underestimation in the model? What about the spatial resolution of the model?

P5 (L167) – 'As expected from the basic chemistry and seen in previous observations' can be replaced by 'In the stratosphere'.

P5 (L169) – 'Underestimating the observed decrease' can possibly be replaced by 'showing higher CO' or something similar.

P6 (L184)- The meaning if this sentence is unclear. So, using analysis and forecast wind fields makes no difference? Does this mean that the wind fields are the same or the results are not sensitive to the winds? And what is the reason for this?

P6 (L193 & L203) – What determines the placement of the plumes in the model? Do models misplace the plumes often?

P7 (Figure 4 description) – It is not easy to compare the data and the model simulations in this figure. I wonder what the differences (data-model, or %) would look like if it's plotted like Fig. 4a and 4b.

P9 (L306) – Full name for GFED version 4 and a citation should be included.

P9 (L312) – 'its relative variability' – what does this mean?

P10 (L320) - A reference is needed for Siberian biomass burning here.

P11 (L353-354) – Is this expected? And why?

P11 (L366-367) - I wonder if and how the MOPITT V6 bias drift is accounted for in this

study.

P12 (L402) – A general website for the ATom mission will be useful to include here.

P12 (references) – Formatting of the references should be corrected to be consistent with the publication standard.

P17 & 18 (Tables 1 & 2) – A map showing all the geographical regions (with specific latitude and longitude) used in Tables 1 & 2 might be useful. I am also curious to know if it would be easier to compare the observations and the model in a form of a bar graph instead of a table (Table 2).

P18 (Fig. 1) – I wonder what would Figs. 1d-1f and Figs 1g-1i would look like if they have the same vertical ranges. The current plots give a false impression that CO is higher between Kona-Pago Pago-Christchurch than the first leg.

---

## Author Comment (AC1) · 11 Jun 2018

The authors use the GEOS-5 chemical forecast system to simulate the CO distribution
and the timing of CO enhancements observed during the Atom-1 aircraft mission in
July-August 2016, using tagged CO tracers to attribute modelled CO to non-biomass
burning and biomass burning (BB) sources from different regions. The authors also
use multi-year satellite measurements (MOPITT and MODIS) of CO and AOD to verify
if August 2016 is representative of typical boreal summer conditions. One of the goals
of Atom is to derive a chemical climatology over the ocean. The paper is well written
and the authors have conveyed their results and findings clearly. The methods used to derive the
results are sound and thorough. I recommend this paper to be published
with some clarification and improvements, detailed below.

**We thank the referee for the positive review and respond to specific comments below.**

Specific comments

1. My major concern is that the authors combined BB emissions from Europe and from
Northern Asia into one tagged CO tracer, which I think leads to difficult attributions of
the simulated North Atlantic CO to emissions in these two regions. One would suspect
that European BB emissions have a more direct impact on North Atlantic CO peaks
than the Asian BB emissions. Moreover, the text seems to imply that BB emissions
from Southern Asia and Australia are not included in the simulation, which might lead
to the underestimation of simulated South Pacific CO peaks.

**We agree that it would be helpful to separate European and northern Asian BB emissions.
However, since we used the GEOS-fp system, we are constrained to use the tagged-CO
region definitions used by the fp system, and the fp system combines European and
northern Asian BB emissions into one tracer. BB emissions from Southern Asia and
Australia are included in the simulation, just not in the Eurasian tracer. They are included
in the "Other BB" tracer instead. We clarify this by adding the following line to the caption
of Table 1: "Other BB does include southern Asia as well Australia." We also added a
supplemental figure that shows the definition of each tagged tracer region on a map.**

2. Can you add to the map showing the Atom flight route the names of locations
mentioned in the text and the labels of flight numbers?

**We added a supplemental figure showing the ATom flight route with locations and flight
numbers labeled.**

3. L116-118: How do you justify adding 20% fossil fuel and 11% BB to CO emissions?
What is the basis, and do you have a reference for the scaling factors? Please also
show the effective CO total emissions, after scaling, in Table 1.

**We include these scalings to account for CO production from VOCs because VOCs are
often emitted along with CO from fossil fuel and BB sources, but the GEOS-fp chemical
mechanism does not carry VOCs and thus does not explicitly calculate the CO production
from these co-emitted VOCs. The reference for these scaling factors is Duncan et al. [2007].
We add the following explanation to the text:**
**"…since VOC's are not explicitly carried in the GEOS-fp chemical mechanism. This
approach was developed by *Duncan et al.* (2007) to account for the CO source from non-**

**methane hydrocarbon oxidation."  We added the effective CO emissions to Table 1 as suggested.**

4. L119-120: Which regions? Methane is well-mixed, and is oxidised in all regions. Could you clarify?

**The regions depend on where oxidation occurs, not where the methane is emitted.  Thus, if methane is oxidized over Asia, the resulting CO is included in the Asian non-BB tracer.  If it is oxidized over North America, it is included in the North American non-BB tracer.  We add the following example to the text:**
**"For example, if methane is oxidized over North America, the resulting CO is included in the North American non-BB tracer."**

5. L152-154: Could you elaborate why the observed peaks in Figure 1d are not captured by the GEOS-5 analysis? Could it be due to the underestimation of Asian BB or/and non-BB in the inventory?

**Tagged tracers in Fig. 1g suggest that they are due to biomass burning, but this could be due to either insufficient emissions or insufficient model resolution.**
**We add the following text to Section 3.1.1: "Either biases in emissions or insufficient vertical or horizontal model resolution may thus be responsible for the underestimate.  The tagged tracer for biomass burning shows a small increase at the time of the underestimated plumes near hour 22 of the Anchorage-Kona flight (Fig. 1d,g), suggesting that those underestimates are due to the insufficient magnitude of the simulated biomass burning plumes."**

6. Did you include biomass burning from South Asia and Australia in your CO simulation? If these are not included, would this lead to the underestimation of the peaks in observed CO in Figure 1f? This is related to the comment above.

**Yes, they are included.  This is now clear from the new supplemental figure showing the tagged tracer region definitions.**

7. L194-196: It is somewhat surprising that this region has notable Asian non-BB influence; could it be the case that the majority of this influence originates from Europe or Central Asia rather than from the Far East?

**Non-BB CO from Europe is tagged separately so would not show up in the Asian non-BB tracer.  We are not able to distinguish Central Asia from East Asia with the tagged tracers. However, the Asian non-BB contribution here, though notable, is not huge, and given the lifetime of CO it is not too surprising that the large Asian CO source would make some contribution to the background even here.**

8. L197-198 and Figure 2f: Would it more precise to attribute the observed CO plumes here to European BB? Could N. American BB contribute to the plumes? Separating European and Asian BB emissions could shed more light on this.

**The tagged tracers show little N. American BB contribution to these plumes.  We agree that separate European and Asian BB tracers would help shed light on this, but as explained above, these are not available.  Examination of the fire emissions from this time period shows fires in both Europe and Asia, making it difficult to separate the two.**

9. L212-213: "An underestimate of Eurasian biomass burning contributes to the model underestimates in the North Pacific and North Atlantic, : : :" – same comment as above.

**As noted above, we do not have the ability to separate the European and Asian biomass burning contributions.**

10. Figures 5 and 6: I suggest to use the same scale between the 2 panels in each figure.

**Done.**

11. L364-369: The negative trends are not necessarily associated with the lower than average value of 2016. Often, IAV is larger than the trend. Whether 2016 is extreme or not should be defined based on statistical metrics.

**We agree that IAV can exceed the trend. The purpose of this discussion is just to point out previous work that shows a trend in MOPITT, and that this can be one factor contributing to the low values in 2016. It is also relevant to the 2[nd] referee's question about MOPITT drift. We add the following clarification:**
**This negative trend may be contributing to the low values in 2016, "although there is also substantial IAV in CO."**

Technical corrections

1. L22: add "boreal" before "summer"

**Done**

2. L293: replace "summer" with "boreal summer/austral winter"

**Done**

3. L335: replace "autumn" with "September to November"

**We now say September-October, since this is when ATom-3 took place.**

4. Figure 8: please tidy up the label on the x-axis.

**Fixed**

5. Figures 11 and 12: please add "August" in the caption.

**Done**

6. Table S1: flights are labelled from 0 to 10 – should it be 1-11?

**Yes, we updated the numbering to be 1-11.**

---

## Author Comment (AC2) · 11 Jun 2018

General Comments

A very unique set of measurements obtained during the ATom-1 mission was analyzed
throughout this study. Comprehensive data sets, both from in-situ and satellites, and
a state-of-the-art global chemistry model were utilized. The analyses methods used
here were reasonable and the results are presented in logical manner. However, it is
not clear to me that what the scientific backgrounds are and what scientific questions
the authors try to answer. What new and unique findings are discussed in this study?
What can we learn from the results presented here? Specifically, why is it important for GEOS-5
to be able to simulate and forecast CO over the ocean? And why ATom-1
(and also the future deployment phases) was planned and how is it different from the
previous campaigns, for instance, HIPPO? I think including more scientific background
and contexts, and specific goals will make this article far more interesting.

**We thank the referee for the thoughtful comments. We have expanded the introduction to
include some of the goals of ATom and relate our modeling work to those goals. In
particular, we now state: "Major goals of the Atom mission include identifying chemical
processes that control the concentrations of short-lived climate forcers, quantifying how
anthropogenic emissions affect chemical reactivity globally, and identifying ways to
improve the modeling of these processes" and "GEOS-5 forecasts help determine the source
regions and emission types that contribute to the trace gas and aerosol concentrations
measured during ATom, which is directly relevant to the goal of quantifying how
anthropogenic emissions affect global chemical reactivity." We also clarify the importance
of validating GEOS-5, adding "GEOS-5 supports numerous aircraft missions, and
validation of the model forecasts is important for developing confidence in and
understanding the limitations of chemistry forecasting for aircraft missions. The ATom
dataset, which uses unbiased sampling rather than chasing plumes, provides a unique
opportunity to validate the overall performance of the GEOS-5 model on a global scale."**

**We also added a supplemental figure showing ta map of the ATom flight tracks.**

**In the future, the ATom Mission PI will prepare an overview paper that contains more
details on the scientific background, planning, goals, and overall accomplishments of ATom.
An extensive discussion of these topics is thus beyond the scope of the present work.**

Specific Comments

P1 (Abstract) – I think the abstract can be rewritten in order for it to serve as a brief
summary of this study. The simplest way is to rearrange it. 1) What is the goal of ATom-
1, 2) What are the observational findings shown in this study, 3) Why is the modeling
study performed and 4) What the conclusion is.

**We reorganized the abstract to begin with a description of ATom and better motivate our
modeling study. We add "The first phase of the Atmospheric Tomography Mission (ATom-
1) took place in July-August of 2016 and included flights above the remote Pacific and
Atlantic oceans. Sampling of atmospheric constituents during these flights is designed to
provide new insights into the chemical reactivity and processes of the remote atmosphere
and how these processes are affected by anthropogenic emissions. Model simulations**

**provide a valuable tool for interpreting these measurements and understanding the origin of the observed trace gases and aerosols, so it is important to quantify model performance" and "We use GEOS-5's tagged tracers for CO to assess the contribution of different emission sources to the regions sampled by ATom-1 to elucidate the dominant anthropogenic influences on different parts of the remote atmosphere".**

P2 (Introduction) – This section can be rewritten to add clarity as well. It should include 1) Scientific background and goals of the ATom-1 (and other phases) mission, 2) What are the scientific questions ATom-1 aimed to answer? 3) What did ATom-1 sample and accomplish? 4) What is the goal of this study?

**We have expanded the introduction as described in the response to General Comments to include information on the goals of ATom and how our work relates to these goals. We also added a link to the ATom website, which includes additional information about ATom.**

In addition, the information in the third and the fourth paragraphs (lines 42-53) provides general overview, instead of specific to this study.

**We reorganized this section to better link the overview to the current study.**

I would also like to know what the ATom-1 specific references are. Is Prather et al. (2017) the only reference relevant to ATom-1?

**We also reference the doi for the ATom dataset. As noted above, an ATom overview paper is planned but not yet published. We added a reference to *Prather et al.* [2018] to Section 5.**

Technical Comments

P3 (Section 2.1) – What are the vertical ranges of ATom-1 measurements? The accuracy or precision of the species measured during ATom-1 should be provided here.

**We state that "Each of the 11 flights included sampling from the boundary layer to the top of the aircraft range (39 kft)." The QCLS observations have an accuracy and precision of 3.5 ppb and 0.15 ppb, respectively. We added that statement to this section.**

P3 (Section 2.2) – The vertical range of MOPITT and MLS and their overlap have to be mentioned specifically. What pressure ranges do they overlap?

**We now clarify that we are using the CO column and 700 hPa CO retrievals from MOPITT and the 215 hPa level for MLS. We also add the following about the overlap: "The MOPITT averaging kernels include some sensitivity to the 200 hPa level, implying a small overlap between the MOPITT and MLS observations."**

How often did ATom-1 sample stratosphere? Are there any ground-based measurements to compare? It has to be mentioned why only the satellite data are used in the comparison. Also, why was the MODIS AOT used?

**We use satellite observations for our analysis of interannual variability because they provide broad spatial coverage over the oceans, where most of the ATom sampling occurred. Ground-based measurements over the oceans are sparse, so we do not include**

them in this analysis.  We use MODIS AOT to examine aerosol interannual variability because MODIS provides a relatively long record.  We added text clarifying these points: **"We focus on satellite observations because they provide broad coverage over the oceans, where surface data is sparse. " "We use MODIS data in this analysis because it provides a relatively long data record."**

P3 (MOPITT) – I believe the current version of MOPITT is 7 not 6.

**This is correct.  We used version 6 in this study because it was a well-established product at the time we conducted this analysis.**

P4 (Section 2.3) – It has to be mentioned why the model is used in this study. What are the questions that can be answered through modeling? Is a global model the sufficient tool for this study?

**We added the purpose of the model to the first sentence, and clarify that a global model is necessary because CO is transported globally.  We added the following text about why we use GEOS-5:**
**"to quantify the contribution of different emission sources to the observed CO distribution and to identify the origin of observed plumes.  A global model is necessary for this analysis since CO is transported globally." and "We use the FP system in our study because it is the system used to generate forecasts that are used during ATom and other aircraft missions, and is thus relevant to future mission and flight planning."**

P4 (L125) – What could be learned from the chemical forecasts during a field campaign, such as, ATom-1? How many pollution or non-pollution events occurred during the ATom-1 period?

**We add a discussion of what chemical forecasts tell us to the first paragraph of Section 3. Pollution plumes are shown in figures 1 and 2. We add this text to Section 3:**
**"The chemical forecasts provide the ATom team with a preview of the chemical environments that the flight is expected to sample, including the location of pollution, biomass burning, or dust plumes; regions of substantial but well-mixed anthropogenic pollution; and cleaner regions.  The forecasts also provide a broader spatial context for the observations, since the 3-dimensional model output shows the spatial extent of features that intersect the flight track."**

P5 (L160) – Is the bias in the emissions responsible for the underestimation in the model? What about the spatial resolution of the model?

**These are both possible explanations.  We add the following clarification:**
**"Either biases in emissions or insufficient vertical or horizontal model resolution may thus be responsible for the underestimate."**

P5 (L167) – 'As expected from the basic chemistry and seen in previous observations' can be replaced by 'In the stratosphere'.

**We now say "As expected from stratospheric chemistry and seen in previous observations"**

P5 (L169) – 'Underestimating the observed decrease' can possibly be replaced by 'showing higher CO' or something similar.

**We changed it to "underestimating the observed gradient" since we want to convey that the change in CO as the flight enters the stratosphere is underestimated.**

P6 (L184)- The meaning if this sentence is unclear. So, using analysis and forecast wind fields makes no difference? Does this mean that the wind fields are the same or the results are not sensitive to the winds? And what is the reason for this?

**It means that the plume timings are already captured by the forecast winds and don't change substantially with the analysis winds. We reword this sentence to say: "Comparison of Fig. S4 with Fig. 2f shows that the impact of using analysis versus forecast wind fields is small for this flight since the forecasts already capture the timing of the plumes."**

P6 (L193 & L203) – What determines the placement of the plumes in the model? Do models misplace the plumes often?

**We added "extent" as well as placement of the plumes. In general, the good agreement in the timing of plumes in the model compared to observations suggests that the model usually places the plumes correctly. However, the placement of plumes is sensitive to vertical and horizontal transport, and to the injection height of the biomass burning emissions. The injection height is a source of uncertainty. In addition, global eulerian models tend to diffuse plumes too quickly. We added the following text to explain this:**
**"The placement and strength of simulated plumes is sensitive to the injection height of the biomass burning, which is a source of uncertainty. In addition, plumes in models tend to dissipate more quickly than in observations due to the numerical effects of limited model resolution (*Eastham and Jacob,* 2017)."**

P7 (Figure 4 description) – It is not easy to compare the data and the model simulations in this figure. I wonder what the differences (data-model, or %) would look like if it's plotted like Fig. 4a and 4b.

**We added a third panel plotting the data − model difference.**

P9 (L306) – Full name for GFED version 4 and a citation should be included.

**We added the full name and the reference to *van der Werf et al.*, [2017]**

P9 (L312) – 'its relative variability' – what does this mean?

**Relative variability means the variability relative to the mean. We add this clarification to the text.**

P10 (L320) - A reference is needed for Siberian biomass burning here.

**We provide references later in this paragraph for large fires in western Russia and Siberia.**

P11 (L353-354) – Is this expected? And why?

**Both CO and aerosols have a major source from biomass burning, so it makes sense that regions influenced by biomass burning, such as the tropical Atlantic, would have high values of both CO and aerosols. We add the following:**

**"This similarity is consistent with the importance of biomass burning emissions for both CO and aerosols."**

P11 (L366-367) - I wonder if and how the MOPITT V6 bias drift is accounted for in this study.

**We do not attempt to account for the MOPITT drift, except to mention it in the text. Since the drift in the CO column is almost negligible [*Deeter et al.*, 2014], we do not expect it to influence our CO column results. There may be a small impact from the drift on the 700 hPa results, but those results are generally consistent with the CO column results.**

P12 (L402) – A general website for the ATom mission will be useful to include here.

**We added a link to the ATom website in the introduction.**

P12 (references) – Formatting of the references should be corrected to be consistent with the publication standard.

**We have formatted the references in this way.**

P17 & 18 (Tables 1 & 2) – A map showing all the geographical regions (with specific latitude and longitude) used in Tables 1 & 2 might be useful. I am also curious to know if it would be easier to compare the observations and the model in a form of a bar graph instead of a table (Table 2).

**We added a supplemental figure showing maps of the geographical regions corresponding to the tagged tracers. We added another supplemental figure showing the ATom flight tracks on a map and labeling the start and end points of the flights, so that the reader can visualize the locations discussed in Table 2. We prefer to keep Table 2 as a table so that we can more easily and quantitatively compare the standard deviations as well as the mean values.**

P18 (Fig. 1) – I wonder what would Figs. 1d-1f and Figs 1g-1i would look like if they have the same vertical ranges. The current plots give a false impression that CO is higher between Kona-Pago Pago-Christchurch than the first leg.

**We considered this, but putting panel e on the same scale as panel d squashes the detail in panel d.**

---

## Referee Report (RR1)

Comments on revision of "Forecasting Carbon Monoxide on a Global Scale for the ATom-1 Aircraft Mission: Insights from Airborne and Satellite Observations and Modeling" by Strode et al.

The authors have sufficiently addressed my comments. However, I would like the authors to consider again the relation between the recent negative trend in CO columns and the lower than average CO values in 2016. It is good to discuss the recent CO trend. But, technically, you cannot make the link, even with a "maybe", between the trend and the value in an individual year. Moreover, in Worden et al. (2013) the trend was calculated over 2000-2012. I suggest the authors to remove the statement "This negative trend may be contributing to the low values in 2016, …" for clarity.

---

## Author Response (AR2)

Comments on revision of "Forecasting Carbon Monoxide on a Global Scale for the ATom-1 Aircraft
Mission: Insights from Airborne and Satellite Observations and Modeling" by Strode et al.

The authors have sufficiently addressed my comments. However, I would like the authors to consider again the relation between the recent negative trend in CO columns and the lower than average CO values in 2016. It is good to discuss the recent CO trend. But, technically, you cannot make the link, even with a "maybe", between the trend and the value in an individual year. Moreover, in Worden et al. (2013) the trend was calculated over 2000-2012. I suggest the authors to
remove the statement "This negative trend may be contributing to the low values in 2016, …" for clarity.

**We thank the referee for the thoughtful comment.  We have removed the statement that "
[revised manuscript text omitted]